# Structure and dynamics of a nanodisc by integrating NMR, SAXS and SANS experiments with molecular dynamics simulations

Tone Bengtsen[1], Viktor L Holm[2], Lisbeth Ravnkilde Kjølbye[3,4], Søren R Midtgaard[2], Nicolai Tidemand Johansen[2], Giulio Tesei[1], Sandro Bottaro[1], Birgit Schiøtt[3], Lise Arleth[2]*, Kresten Lindorff-Larsen[1]*

[1]Structural Biology and NMR Laboratory and Linderstrøm-Lang Centre for Protein Science, Department of Biology, University of Copenhagen, Copenhagen, Denmark; [2]Structural Biophysics, X-ray and Neutron Science, Niels Bohr Institute, University of Copenhagen, Copenhagen, Denmark; [3]Department of Chemistry, Aarhus University, Aarhus, Denmark; [4]Novo Nordisk A/S, Måløv, Denmark

**Abstract** Nanodiscs are membrane mimetics that consist of a protein belt surrounding a lipid bilayer, and are broadly used for characterization of membrane proteins. Here, we investigate the structure, dynamics and biophysical properties of two small nanodiscs, MSP1D1ΔH5 and ΔH4H5. We combine our SAXS and SANS experiments with molecular dynamics simulations and previously obtained NMR and EPR data to derive and validate a conformational ensemble that represents the structure and dynamics of the nanodisc. We find that it displays conformational heterogeneity with various elliptical shapes, and with substantial differences in lipid ordering in the centre and rim of the discs. Together, our results reconcile previous apparently conflicting observations about the shape of nanodiscs, and pave the way for future integrative studies of larger complex systems such as membrane proteins embedded in nanodiscs.

*For correspondence:
arleth@nbi.ku.dk (LA);
lindorff@bio.ku.dk (KL-L)

Competing interests: The authors declare that no competing interests exist.

## Introduction

Nanodiscs are widely used membrane models that facilitate biophysical studies of membrane proteins (*Bayburt et al., 2002*). They are derived from, and very similar to, the human ApoA1 protein from high density lipoproteins (HDL particles) and consist of two amphipatic membrane scaffold proteins (MSPs) that stack and encircle a small patch of lipids in a membrane bilayer to form a discoidal assembly. The popularity of nanodiscs arises from their ability to mimic a membrane while at the same time ensuring a small system of homogeneous composition, the size of which can be controlled and can give diameters in a range from about 7 to 13 nm (*Denisov et al., 2004*; *Hagn et al., 2013*).

Despite the importance of nanodiscs in structural biology research and the medical importance of HDL particles, we still lack detailed structural models of these protein-lipid particles. The nanodisc has so far failed to crystallize, so a range of different biophysical methods have been used to provide information about specific characteristics. For example, mass spectrometry experiments have provided insight into lipid-water interactions and heterogeneous lipid compositions (*Marty et al., 2014*; *Marty et al., 2015*), solid state NMR has been used to quantify lipid phase transition states and lipid order (*Mörs et al., 2013*; *Martinez et al., 2017*) and small angle X-ray scattering (SAXS) and -neutron scattering (SANS) have provided insight into the size and low resolution shape of nanodiscs in solution (*Denisov et al., 2004*; *Skar-Gislinge et al., 2010*; *Midtgaard et al., 2014*). These

experiments have been complemented by molecular dynamics (MD) simulations that provided both pioneering insights into the structure (*Shih et al., 2005*; *Shih et al., 2007*) as well as a better understanding of the assembly process, lipid-protein interactions and how much a nanodisc mimics the membrane bilayer (*Siuda and Tieleman, 2015*; *Debnath and Schäfer, 2015*; *Vestergaard et al., 2015*).

A high resolution structure of the MSP protein belt encircling the nanodisc was recently obtained from the small, helix-5-deleted nanodisc, MSP1D1ΔH5 (henceforth ΔH5), reconstituted with DMPC lipids (ΔH5-DMPC) (*Bibow et al., 2017*) by combining nuclear magnetic resonance (NMR) spectroscopy, electron paramagnetic resonance (EPR) spectroscopy and transmission electron microscopy (TEM) (*Bibow et al., 2017*). While these experiments were performed on lipid-loaded nanodiscs, the study focused on the protein components, and on determining a time- and ensemble averaged structure of these, but left open the question of the role of the lipids (*Martinez et al., 2017*) as well as any structural dynamics of the overall nanodisc. Intriguingly, the resulting structure of the belt proteins corresponded to that of an almost circularly-shaped disc, while our previous SAXS/SANS investigations are clearly consistent with discs with an on-average elliptical cross-section (*Skar-Gislinge et al., 2010*; *Midtgaard et al., 2015*).

Here, we build upon this work to study the structure and dynamics of the nanodisc and the lipid properties in the disc. We performed SAXS and SANS experiments on the ΔH5-DMPC variant, and integrated these with MD simulations and the NMR data (*Bibow et al., 2017*) through an integrative Bayesian/maximum entropy (BME) approach (*Hummer and Köfinger, 2015*; *Różycki et al., 2011*; *Bottaro et al., 2020*; *Bottaro et al., 2018*; *Orioli et al., 2020*). We thereby obtain a model of the conformational ensemble of the ΔH5-DMPC nanodisc that is consistent with the structural information obtained from each method, as well as our molecular simulations, and which successfully explains differences in previous structural interpretations. In addition, we study the lipid ordering in our ensemble, and use the results to aid in the interpretation of Differential Scanning Calorimetry (DSC) measurements of the melting transition of DMPC in differently sized nanodiscs. Our study exemplifies how these integrative methods can be used to protein-lipid systems, possibly paving the way for future studies of membrane proteins embedded in nanodiscs.

## Results and discussion

### Structural investigations of ΔH5-DMPC and ΔH4H5-DMPC nanodiscs by SAXS and SANS

We determined optimal reconstitution ratios between the DMPC lipids and the ΔH5 and ΔH4H5 protein belts to form lipid-saturated nanodiscs based on a size-exclusion chromatography (SEC) analysis (*Figure 1—figure supplement 1* and Materials and methods). In line with previous studies (*Hagn et al., 2013*), we found that reconstitution ratios of 1:33 for ΔH4H5:DMPC and 1:50 for ΔH5: DMPC were optimal in order to form single and relatively narrow symmetric peaks. Building upon earlier work for other discs (*Denisov et al., 2004*; *Skar-Gislinge et al., 2010*) we performed combined SEC-SAXS and SEC-SANS experiments to determine the size and shape of DMPC loaded ΔH5 and ΔH4H5 nanodiscs (*Figure 1*). These experiments were performed at 10°C, and based on results from previous NMR experiments on nanodiscs (*Martinez et al., 2017*) as well as a melting temperature $T_M \approx 24$°C for DMPC, where we expect the lipids to be in the gel-phase. Our SAXS and SANS data all exhibit a flat Guinier region at low $q$ and indicate no signs of aggregation (*Figure 1A,B*). In both the ΔH5-DMPC and ΔH4H5-DMPC systems, the SAXS data exhibit an oscillation at medium to high $q$ ([0.05:0.2] Å⁻¹) arising from the combination of a negative excess scattering length density of the hydrophobic alkyl-chain bilayer core and positive excess scattering length densities of the hydrophilic lipid PC-headgroups and the amphipathic protein belt. The SANS data decreases monotonically as a function of $q$ in accordance with the homogeneous contrast situation present here. These two different contrast situations, core-shell-contrast for SAXS and bulk-contrast for SANS, are also clearly reflected in the obtained $p$(r)-functions (*Figure 1C,D*), which also confirm that the ΔH5-DMPC nanodiscs are slightly larger than the ΔH4H5-DMPC nanodiscs.

Our data are in qualitative agreement with the SAXS and SANS data obtained for MSP1D1 nanodiscs (*Denisov et al., 2004*; *Skar-Gislinge et al., 2010*) and similar systems (*Midtgaard et al., 2014*; *Midtgaard et al., 2015*), and indicate an 'on average' discoidal structure. Therefore, w first analyzed

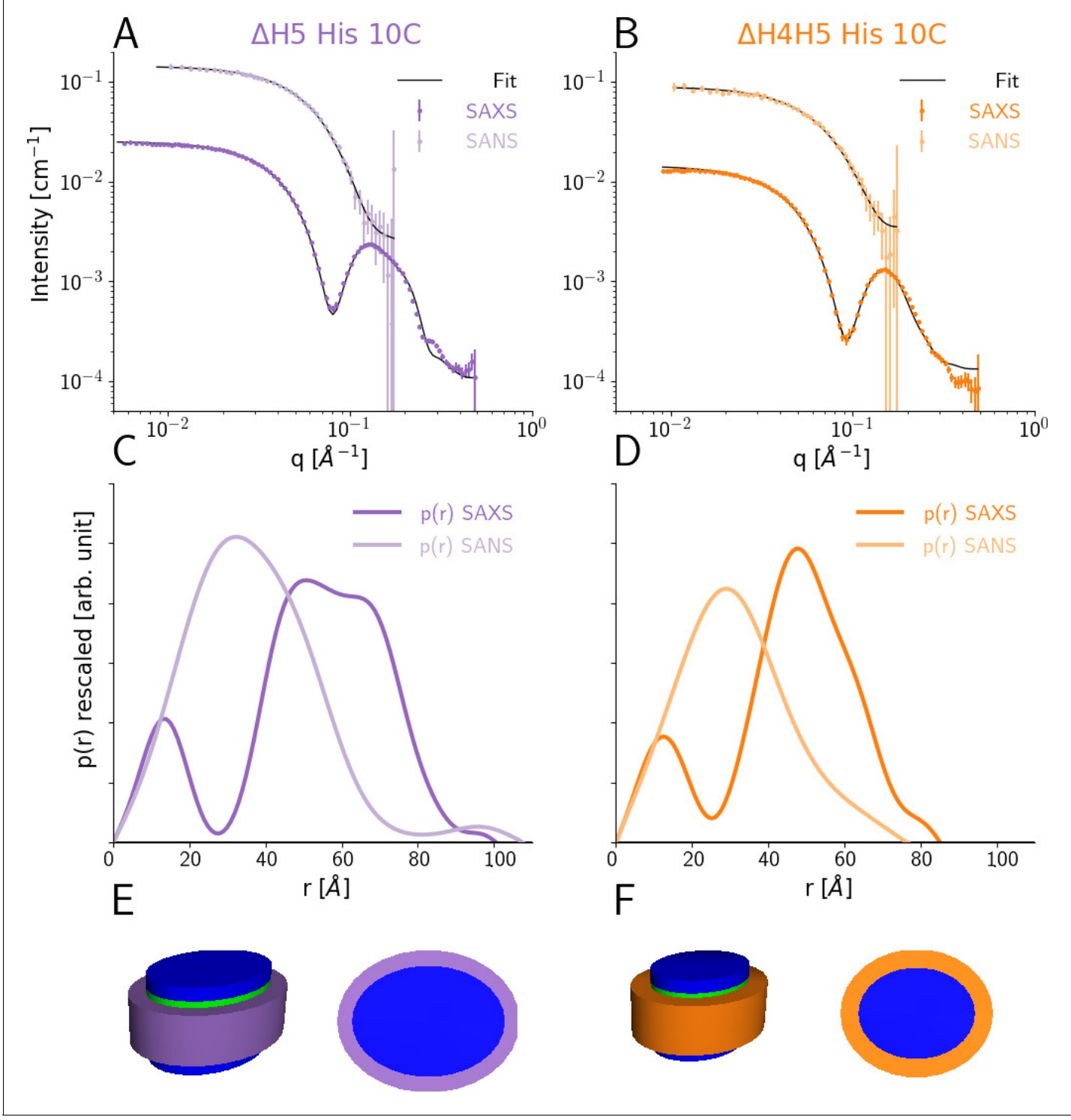

**Figure 1.** SEC-SAXS and SEC-SANS analysis of nanodiscs. (**A**) SEC-SAXS (dark purple) and SEC-SANS (light purple) for ΔH5-DMPC nanodiscs at 10°C. The continuous curve show the model fit corresponding to the geometric nanodisc model shown in E. (**B**) SEC-SAXS (dark orange) and SEC-SANS (light orange) data for the ΔH4H5-DMPC nanodiscs at 10°C. (**C,D**) Corresponding pair-distance distribution functions. (**E, F**) Fitted geometrical models for the respective nanodiscs (drawn to scale relative to one another).

The online version of this article includes the following figure supplement(s) for figure 1:

**Figure supplement 1.** SEC analysis of the reconstitution of ΔH4H5 and ΔH5 nanodiscs with DMPC.

**Figure supplement 2.** Model-based interpretation of the SAXS/SANS data on DMPC based nanodiscs obtained under different conditions.

*Figure 1 continued on next page*

**Figure supplement 3.** Varying the axis ratio in the model.
**Figure supplement 4.** Introducing polydispersity in the model.

the scattering data by global fitting of a previously developed molecular-constrained geometrical model for the nanodiscs (*Skar-Gislinge et al., 2010*; *Skar-Gislinge and Arleth, 2011*; *Pedersen et al., 2013*). The model (see Materials and methods) describes the lipid interior of the nanodisc as a stack of flat, elliptically-shaped discs that consists of the hydrophobic bilayer that is sandwiched in between the two hydrophilic headgroup layers. The inner lipid part of the nanodisc is encircled by a hollow cylinder with an elliptical cross-section, which models the two protein MSP-belts stacked upon one another (*Figure 1E,F*). Using this model, we obtained excellent simultaneous fits to SAXS and SANS data for both the ΔH4H5-DMPC and ΔH5-DMPC nanodiscs (*Figure 1A,B*).

We find the area per headgroup, $A_{head}$, for DMPC for both systems (ca. 55 Å$^2$ ; *Table 1* left), somewhat higher than the $A_{head}$ of gel-phase DMPC (47.2 ± 0.5 Å$^2$ at 10°C) (*Tristram-Nagle et al., 2002*), but in agreement with the very broad melting transition observed in our DSC data (see below). We find 65 ± 13 and 100 ± 14 DMPC molecules in the nanodiscs for ΔH4H5 and ΔH5, respectively, in agreement with the reconstitution ratios reported above.

## Temperature dependence probed by SAXS and SANS

We continued to investigate the impact of temperature and His-tags on both the SAXS measurements and the resulting geometrical model of ΔH5-DMPC. We acquired standard solution SAXS data for a new preparation of the ΔH5-DMPC nanodiscs, this time without His-tags and measured at both 10°C and 30°C. At these two temperatures the DMPC is expected to be dominantly in the gel

**Table 1.** Parameters of the SAXS and SANS model fit.

Left: Parameters for the simultaneous model fits to SEC-SAXS and SEC-SANS of His-tagged nanodiscs (denoted -His) for both ΔH4H5-DMPC and ΔH5-DMPC. Both measurements were obtained at 10°C. Right: Standard solution SAXS measurements of the ΔH5-DMPC nanodisc without His-tags (denoted -ΔHis) obtained at two different temperatures, in the gel phase at 10°C and in the liquid phase at 30°C. * marks parameters kept constant.

| | SEC-SAXS+SEC-SANS | | SAXS | |
|---|---|---|---|---|
| | ΔH4H5-His | ΔH5-His | ΔH5-ΔHis | ΔH5-ΔHis |
| $T$ | 10°C | 10°C | 10°C | 30°C |
| $\chi^2_{reduced}$ | 1.95 | 5.12 | 3.76 | 2.40 |
| | Fitting Parameters | | | |
| Axis Ratio | 1.3 ± 0.4 | 1.2 ± 0.2 | 1.4 ± 0.1 | 1.3 ± 0.1 |
| $A_{Head}$ | 55 ± 5 Å$^2$ | 54 ± 2 Å$^2$ | 52 ± 2 Å$^2$ | 60 ± 3 |
| $H_{Belt}$ | 24* Å$^2$ | 24* Å$^2$ | 24* Å$^2$ | 24* Å$^2$ |
| $N_{Lipid}$ | 65 ± 13 | 100 ± 14 | 102 ± 7 | 104 ± 9 |
| $CV_{belt}$ | 1* | 1* | 1* | 0.97 ± 0.02 |
| $CV_{lipid}$ | 1.00 ± 0.02 | 1.01 ± 0.01 | 1.003 ± 0.007 | 1.044 ± 0.007 |
| $Scale_{x-ray}$ | 1.13 ± 0.28 | 1.1 ± 0.2 | 1.2 ± 0.1 | 1.2 ± 0.2 |
| $Scale_{neutron}$ | 1.7 ± 0.5 | 0.8 ± 0.2 | - | - |
| | Results From Fits | | | |
| $H_{lipid}$ | 40 Å | 41 Å | 41 Å | 38 Å |
| $H_{tails}$ | 28 Å | 28 Å | 29 Å | 26 Å |
| $R_{major}$ | 27 Å | 32 Å | 34 Å | 36 Å |
| $R_{minor}$ | 21 Å | 27 Å | 25 Å | 28 Å |
| $W_{belt}$ | 10 Å | 9 Å | 9 Å | 9 Å |

and liquid phase, respectively, as they are below and above the melting transition temperature (*Martinez et al., 2017*) (see also DSC analysis below). We used a standard solution SAXS setup for these measurements, as this at present provides a better control of both the sample temperature and sample concentration than in the SEC-SAXS based measurement. The effect of the DMPC melting transition is clearly reflected in the SAXS data (*Figure 1—figure supplement 2*) where both the position of the first minimum and the shape of the oscillation changes as the DMPC transitions from the gel to the molten state. We observe that the intensity of the forward scattering decreases significantly with increasing temperature, a result of the small but significant temperature-dependent change of the partial specific molecular volume of the DMPC.

To analyze the data, we again applied the molecular constrained geometrical model for the nanodiscs (*Table 1*, Right). Here, the effect of the DMPC melting transition can clearly be seen on the obtained DMPC area per headgroup which increases significantly as a result of the melting. Qualitatively similar observations of the melting transition of DMPC and DPPC based nanodiscs were previously reported in the MSP1D1 and MSP1E3D1 nanodiscs using DSC, SAXS and fluorescence (*Denisov et al., 2005*; *Graziano et al., 2018*). Regarding the shape of the ΔH5 nanodiscs without the His-tag (*Figure 1—figure supplement 2*), we find parameters similar to those derived from SEC-SAXS/SANS experiments including an elliptical shape with ratios of the two axes between 1.2 and 1.4. This observation is in apparent contrast to the recently described integrative NMR/EPR structural model of the ΔH5-DMPC nanodisc which was found to be more circular (*Bibow et al., 2017*). We therefore examined the fit to the model varying the axis ratios from 1.0 to 1.6 and indeed find that a number of features are best explained with a slightly asymmetric model (*Figure 1—figure supplement 3*). Both in the SEC-SAXS/SANS experiments, but perhaps particularly in the standard solution SAXS setup, it is possible that polydispersity in the number of lipids embedded in the nanodiscs is present (*Skar-Gislinge et al., 2018*), and contributes to the shapes obtained from our models (*Caponetti et al., 1993*). We therefore analyzed our data using a model where we include polydispersity through a normally-distributed number of lipids, parameterized via the relative standard deviation ($\sigma_{lip}$). Our results show that while a modest level of polydispersity (ca. 1%) cannot be ruled out, greater levels lead to worsening of the fit to the data (*Figure 1—figure supplement 4*).

## Molecular dynamics simulations

The results described above suggest an apparent discrepancy of the solution structure of the ΔH5-DMPC nanodisc when viewed either by NMR/EPR or SAXS/SANS. In particular, the NMR/EPR structure revealed a circular shape whereas the SAXS/SANS experiments suggested an elliptical shape. The two kinds of experiments, however, differ substantially in the aspects of the structure that they are sensitive to. Further, both sets of models were derived in a way to represent the distribution of conformations in the experiments by a single 'average' structure.

In order to understand the structural discrepancies between the two solution methods better, and to include effects of conformational averaging, we performed atomistic MD simulations of the His-tag truncated ΔH5-DMPC nanodisc. In these simulations, we mimicked the experimental conditions of the standard solution SAXS measurements obtained at 30°C and used 100 DMPC lipids in the bilayer as found above. We performed two simulations (total simulation time of 1196 ns) using the CHARMM36m force field (*Huang et al., 2017*). We visualized the conformational ensemble of the ΔH5-DMPC nanodisc by clustering the simulations, and found that the three most populated clusters represent 95% of the simulations. Notably, these structures all have elliptical shapes, but differ in the directions of the major axis (*Figure 2A*).

We then examined the extent to which the simulations agree with the ensemble-averaged experimental data, focusing on the SAXS experiments and NOE-derived distance information from NMR. We calculated the SAXS intensities from the simulation frames using both FOXS (*Schneidman-Duhovny et al., 2013*; *Schneidman-Duhovny et al., 2016*, *Figure 2B*) and CRYSOL (*Svergun et al., 1995*, *Figure 2—figure supplement 1*) and compared to the corresponding standard solution SAXS experiments obtained at 30°C. Similarly, we used $r^{-3}$-weighted averaging to calculate the effective distances in the simulations and compared them to the previously reported methyl (*Figure 2C*) and amide NOEs (*Figure 2—figure supplement 2*, *Bibow et al., 2017*). The discrepancy observed between the simulation and the experiments were quantified by calculating $\chi^2$ (*Table 2*).

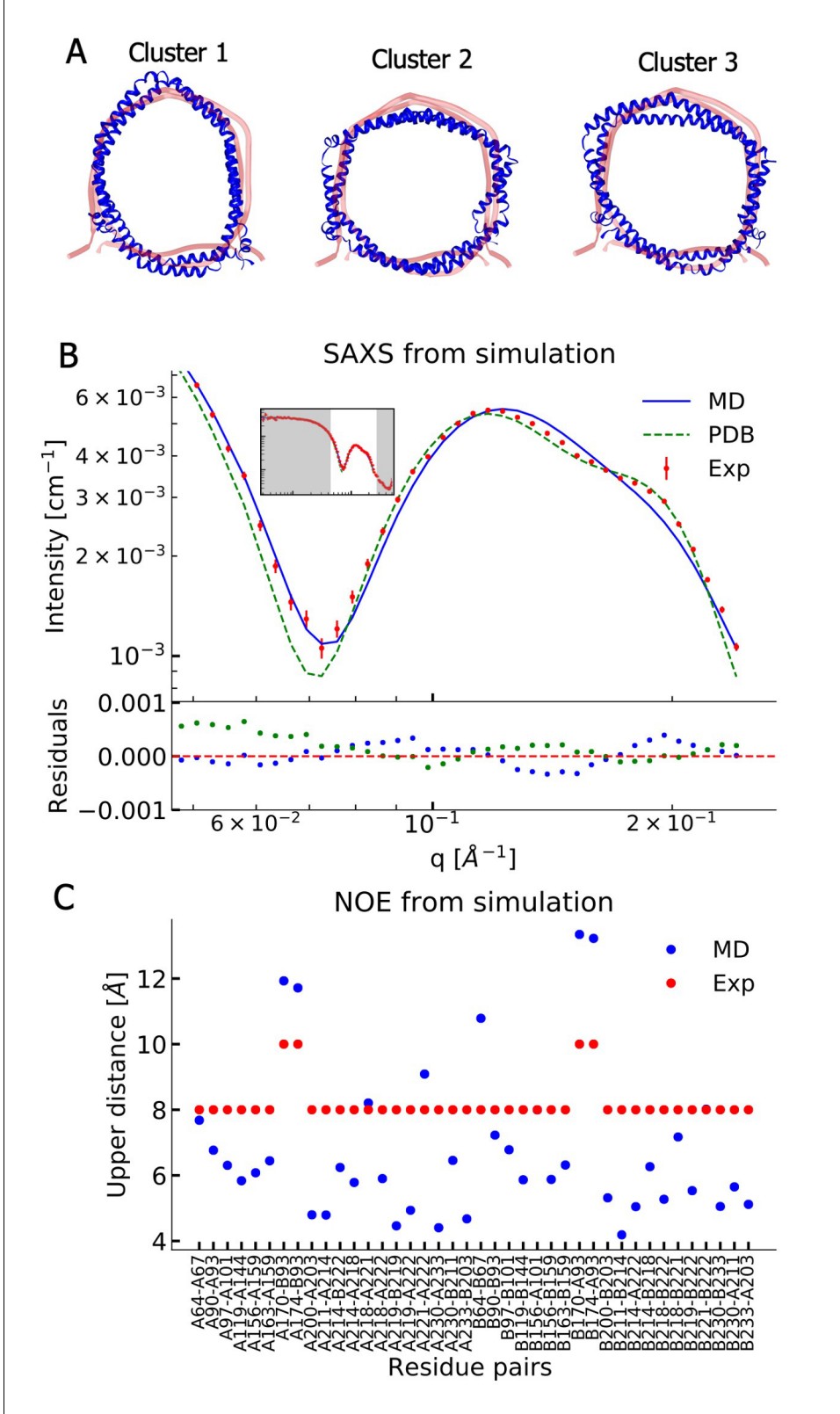

**Figure 2.** Comparing MD simulations with experiments. (A) Visualization of the conformational ensemble from the MD simulation by clustering (blue). Only the protein parts of the nanodisc are visualized while the lipids are left out to emphasize the shape. The top three clusters contain 95% of all frames. The previous NMR/EPR-structure is shown for comparison (red). (B) Comparison of experimental standard solution SAXS data (red) and SAXS calculated from the simulation (blue). Green dotted line is the back-calculated SAXS from the integrative NMR/EPR-structure (labelled PDB). Residuals

*Figure 2 continued on next page*

*Figure 2 continued*

for the calculated SAXS curves are shown below. Only the high *q*-range is shown as the discrepancy between simulation and experiments are mainly located here (for the entire q-range see *Figure 2—figure supplement 1*). (C) Comparison of average distances from simulations (blue) to upper-bound distance measurements (red) between methyl NOEs. The labels show the residues which the atoms of the NOEs belong to.

The online version of this article includes the following figure supplement(s) for figure 2:

**Figure supplement 1.** Comparing simulations with SAXS data.
**Figure supplement 2.** HN-NOE.
**Figure supplement 3.** Example of a His-tagged nanodisc used for SANS calculations.
**Figure supplement 4.** Comparing MD simulations with SANS data.
**Figure supplement 5.** Comparing MD simulations with PRE data.
**Figure supplement 6.** Comparing MD simulations with EPR data.

The comparison between experiments and simulations reveal an overall good agreement between the two. Interestingly, the simulations agree well with the SAXS data in the *q*-region where scattering is dominated by the lipid bilayer and where our geometric fitting of the models for SAXS generally are very sensitive. The MD simulation trajectory captures accurately the depth of the SAXS minimum around $q = 0.07$ Å$^{-1}$; however, the shoulder observed in the experiments in the range 0.15 Å$^{-1}$–0.20 Å$^{-1}$ is not captured accurately.

Direct comparison of the previously determined integrative NMR/EPR structure (*Bibow et al., 2017*) to the SAXS data is made difficult by the missing lipids in the structure. We thus built a model of the lipidated structure by first adding DMPC lipids to the NMR/EPR solved structure (PDB ID 2N5E), and then equilibrating only the lipids by MD, keeping the protein conformation fixed. When we use this structure to calculate the SAXS data, the back-calculated data overshoots the depth of the SAXS minimum but captures well the shoulder observed in the experimental data (*Figure 2B*). Thus, neither the MD trajectory nor the NMR/EPR structure fit perfectly with the measured SAXS data.

When comparing the simulations to the NMR-derived distances between methyl groups (*Figure 2C*), we generally find good agreement, but observe a few distances that exceed the experimental upper bounds. A similar trend is observed in the comparison to amide NOEs (*Figure 2—figure supplement 2*) which shows overall good agreement but with a few NOEs violating at similar positions as for the methyl NOEs. As the amide NOEs are mostly sensitive to the local helical structure, the good agreement with this data mostly reflects that the secondary structures are maintained during the simulations.

We also compared the simulations to the SANS data for ΔH5-DMPC. The scattering contrast is very different in SAXS and SANS, and the scattering from the lipid bilayer has a relatively higher amplitude in the latter. This gives an independent check that the simulation provides a good description of the structure of the lipid bilayer. As the SEC-SANS data were measured on a His-tagged ΔH5-DMPC nanodisc, we therefore simulated this situation by creating an ensemble of His-

**Table 2.** Comparing experiments and simulations.
We quantify agreement between SAXS and NMR NOE experiments by calculating the $\chi^2$. The previously determined NMR structure (*Bibow et al., 2017*) (PDB ID 2N5E) is labelled PDB, the unbiased MD simulation by MD, and simulations reweighted by experiments are labelled by MD and the experiments used in reweighting. $S_{rel}$ is a measure of the amount of reweighting used to fit the data (*Bottaro et al., 2018*) (see Methods for more details).

| Data for integration | $S_{rel}$ | $\chi^2$ | |
|---|---|---|---|
| | | SAXS | NOE |
| PDB | – | 2.9 | 9.5 |
| MD | 0 | 10.0 | 8.2 |
| MD + SAXS | -1.7 | 1.5 | 7.9 |
| MD + NOE | -1.9 | 8.9 | 4.2 |
| MD + SAXS + NOE | -1.7 | 1.9 | 6.0 |

tag structures and randomly sampled and attached these to the outer MSP-belts in the simulation frames under the assumption that the His-tags are disordered on the nanodiscs (*Figure 2—figure supplement 3*). As for the SAXS and NOE data we also here find a generally good agreement (*Figure 2—figure supplement 4*).

As a final consistency check, we compared our simulations to NMR paramagnetic relaxation enhancement (PRE) data (*Figure 2—figure supplement 5*) and EPR data (*Figure 2—figure supplement 6*), that both use spin-labels to probe longer range distances. As reference, we used the calculation of the PRE and EPR data from the structure that was derived using these and the remaining NMR data (*Bibow et al., 2017*) and find comparable agreement.

## Integrating experiments and simulations

While the MD simulations are overall in good agreement with the SAXS and NMR NOE data, there remain discrepancies that could contain information about the conformational ensemble of ΔH5-DMPC in solution. We therefore used a previously described Bayesian/Maximum Entropy (BME) approach (*Hummer and Köfinger, 2015*; *Różycki et al., 2011*; *Bottaro et al., 2018*; *Cesari et al., 2016*; *Bottaro et al., 2020*) to integrate the MD simulations with the SAXS and NMR data. Briefly, the BME method refines the simulation against measured experimental averages by balancing (1) minimizing the discrepancy between the simulation and the observed experimental averages and (2) ensuring as little perturbation of the original simulation as possible thereby limiting chances of over-fitting. The outcome is a conformational ensemble that is more likely to represent ΔH5-DMPC in solution. In practice, this is achieved by changing the weight of each configuration in the ensemble obtained from the MD simulations, and we therefore call this a 'reweighted ensemble' (*Bottaro and Lindorff-Larsen, 2018*; *Bottaro et al., 2020*). The amount of reweighting can be quantified by an entropy change ($S_{rel}$) that reports on how much the weights had to be changed to fit the data (*Bottaro et al., 2018*) (see Materials and methods). Alternatively, the value $\phi_{eff} = \exp(S_{rel})$ reports on the effective ensemble size, that is what fraction of the original frames that were used to derive the final ensemble (*Orioli et al., 2020*). We note that we reweight each individual conformation in the ensemble, and thus that the clustering is only used for presenting the results. In this way we avoid uncertainties that come from difficulties in clustering heterogeneous ensembles.

We used both the SAXS and NOE data individually, as well as combined, to understand the effects of each source of data on the reweighted conformational ensemble (*Table 2*). We note that when a specific type of data is used to generate the ensemble, the resulting $\chi^2$ simply reports on how well the simulation has been fitted to the data; because of the maximum-entropy regularization to avoid overfitting, we do not fit the data as accurately as possible. The two types of experimental data complement each other in structural information content. Specifically, the SAXS data report on the overall size and shape, and is sensitive to both protein and lipids through atom-atom pair distributions in a range starting from ≈ 10 Å, whereas the NOEs contain local, specific atom-atom distances from the protein belts of the ΔH5 but not any direct information about the lipids.

We find that refining against a single of the two data types only improves the MD trajectory with respect to the structural properties it is sensitive to, highlighting the orthogonal information in the two sources of information. In addition, we performed reweighting with the methyl NOEs and the amide NOEs separately (*Figure 3—figure supplement 4*). The already low discrepancy of the amide NOEs barely improves while the discrepancies of both methyl NOEs and SAXS are unaffected by integration with amide NOEs alone, implying that the structural information content contained in the amide NOEs (mostly secondary structure) is already correctly captured by the force field and starting structure. Because the NOE and SAXS experiments provide independent information we refined the ensemble against both sets of data (*Figure 3*). We find that we can fit both sources of data at reasonable accuracy without dramatic changes of the weights away from the Boltzmann distribution of the force field ($\phi_{eff} = 18\%$).

Finally, we used the PRE and EPR data to validate the refined ensemble. In general we find comparable and overall good agreement between the original NMR/EPR structure and our MD refined ensembles, suggesting that our ensembles are in good agreement with data that was not used directly as input in the refinement (*Figure 2—figure supplement 5* and *Figure 2—figure supplement 6*). We further find that reweighting the MD simulations against the SAXS and NOE data generally improves the agreement to the EPR data. We thus proceed with our analysis of the structural

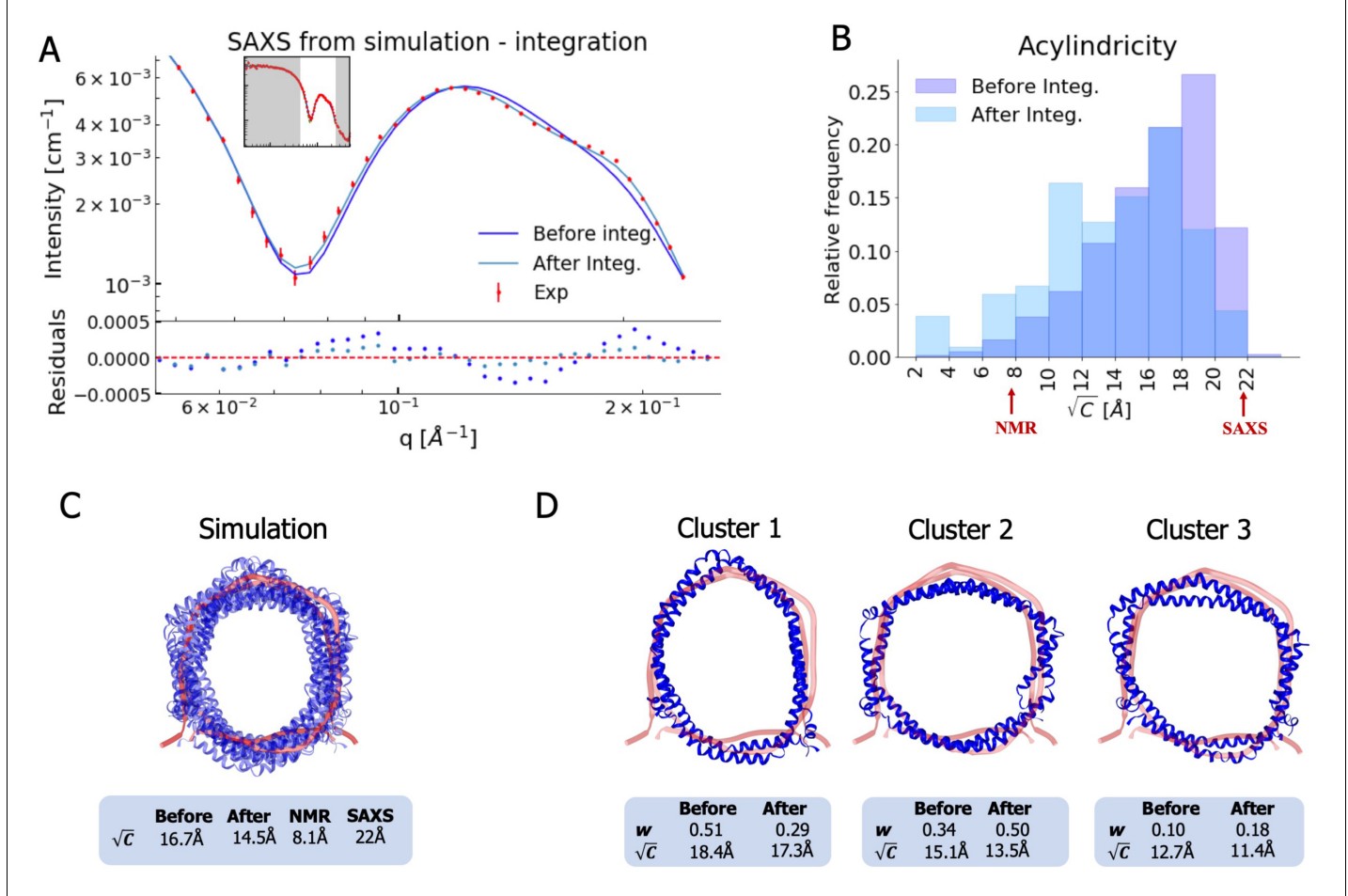

**Figure 3.** Integrating simulations and experiments. (**A**) SAXS data calculated from the simulation before and after reweighting the ensemble using experimental data. Only the high $q$-range is shown as the discrepancy between simulation and experiments are mainly located here (for the entire $q$-range see *Figure 3—figure supplement 1*). Agreement with the NOEs before and after integration are likewise shown in *Figure 3—figure supplement 2*. (**B**). Histogram of the acylindricity of the simulations ($\sqrt{C}$) both before integration (dark blue) and after integration (light blue). (**C**) Visualization of the conformational ensemble showing structures sampled every 100 ns in cartoon representation (blue), the original NMR/EPR structure is shown in rope representation for comparison (red). The table below shows the acylindricity of the entire conformational ensemble before and after integration and compared to the original NMR/EPR (NMR) structure and the SAXS/SANS model fit. (**D**) Weights and acylindricity of the three main clusters of the MD simulation (blue) before and after integration.

The online version of this article includes the following figure supplement(s) for figure 3:

**Figure supplement 1.** Comparison of the experimental SAXS data from simulation before and after integration.

**Figure supplement 2.** NOEs from simulation before and after reweighting.

**Figure supplement 3.** Determination of θ.

**Figure supplement 4.** Combining experiments and simulations.

features of ΔH5-DMPC using an ensemble of conformations that is based on integrating the MD simulations with both the SAXS and NOE experiments.

Analysis of the measured SAXS and SANS revealed an elliptical shape of the ΔH5-DMPC upon fitting of a single structure to the data. In contrast, the structure obtained by fitting the NMR/EPR data to a single structure gave rise to a more circular configuration. Combining the results of the two studies, we hypothesized that the nanodisc possesses underlying elliptical fluctuations with the major axis changing within the nanodisc. In such a system NMR and EPR measurements, which build on ensemble averaged information of specific atom-atom interactions, will give rise to an on-average circular structure. SAXS and SANS, on the contrary, which build on distributions of global distances rather than specific atom-atom distances, will not distinguish between the orientation of the major

axis within the nanodisc and thus give rise to observations of an elliptical shape. By complementing the experiments with MD simulations we obtained a ensemble with structural features that support this hypothesis.

We thus quantified the degree of ellipticity in terms of an acylindricity parameter, $C$, defined as the difference between the $x$ and $y$ components of the gyration tensor (see Materials and methods for details). $C$ is thus a measure of how far from a perfect circular cylinder the shape is, and $C = 0$ corresponds to a circular shape. We calculated both the average and distribution of the acylindricity from the simulated ensemble both before and after reweighting against the experimental data (*Figure 3B and C*). In addition, we calculated the acylindricity of both the integrative NMR structure and from the structural model obtained from the SAXS and SANS measurements.

We find that the acylindricity decreased from $\sqrt{C} = 17$ Å in the original MD simulation trajectory to $\sqrt{C} = 15$ Å after integration of the NMR and SAXS data, showing that the experiments indeed affect the structural features. This value is in the middle of that obtained from the analytical geometric model fitted to the SAXS data ($\sqrt{C} = 22$ Å) and that of the integrative NMR/EPR structure ($\sqrt{C} = 8$ Å) (*Bibow et al., 2017*). Thus, the acylindricity of the final, heterogenous ensemble lies between that of the two conformations that were fitted as single structures to fit either the NMR or SAXS data.

To understand better the elliptical shape of the ΔH5-DMPC nanodisc and the role played by reweighting against experiments, we calculated the average acylindricity for each cluster of conformations of ΔH5-DMPC both before and after integration with experimental data (*Figure 3D*). We note that because our reweighting procedure acts on the individual conformations and not at the coarser level of clusters, the average acylindricity changes slightly for each cluster upon reweighting. Clusters 1 and 2, which together constitute about 80% of the conformational ensemble (both before and after reweighting), are both clusters with high acylindricity, but with almost orthogonal directions of the major axis in the elliptical structure. The major change after integration is the exchange in populations of the two clusters resulting in cluster 2 to be weighted highest, underlining the influence and importance of the integration. Thus, our MD simulations and the integration with the experiments support the hypothesis of underlying elliptical fluctuations with the major axis changing direction inside the nanodisc, and we note that the detailed molecular description of this was only possible by combining the MD simulations with both the SAXS and NMR data.

## Analyses of the lipid properties in nanodiscs

Nanodiscs are often used as models for extended lipid bilayers, but the presence and interactions with the protein belt — and the observed shape fluctuations — could impact the properties of the lipid molecules in the nanodisc compared to a standard bilayer. Building on earlier experimental (*Mörs et al., 2013*; *Martinez et al., 2017*) and simulation work (*Siuda and Tieleman, 2015*; *Debnath and Schäfer, 2015*) work, we therefore used our experimentally-derived ensemble of nanodisc structures to investigate the properties of lipids in the small ΔH5-DMPC nanodisc, and compared them to those in a DMPC bilayer. Specifically, we calculated the thickness of the DMPC bilayer (*Figure 4A*) and the order parameters, $S_{CH}$, of the DMPC lipids (*Figure 4B,C*).

As done previously (*Siuda and Tieleman, 2015*; *Debnath and Schäfer, 2015*), we subdivide the lipid area in the nanodisc into zones dependent on the distance from the MSP protein belts (above or below 10 Å). The results of both the thickness and order parameter analyses show the same trend: a clear difference between the lipids close to the protein belt and those more central in the nanodisc. The results illustrates that the DMPC lipid bilayer in the ΔH5 nanodiscs is not homogeneous but rather thinner and un-ordered near the protein belt and thicker and more ordered in the core of the nanodisc, which in turn is more similar to a pure bilayer (*Figure 4*). These results are in line with previous simulation studies on the larger DMPC nanodiscs, MSP1, MSP1E1 and MSP1E2 (*Siuda and Tieleman, 2015*; *Debnath and Schäfer, 2015*), albeit performed without experimental reweighting, as well as with solid state NMR data on the both ΔH5-DMPC and the larger MSP1-DMPC (*Mörs et al., 2013*; *Martinez et al., 2017*).

We proceeded by performing DSC experiments on nanodiscs of different sizes to examine the impact of the differentiated lipid order in the core and rim of the nanodisc. Specifically, we examined the lipid melting transition of DMPC inside ΔH4H5, ΔH5 and the larger MSP1D1 nanodiscs, and used pure DMPC vesicles as reference. In line with earlier DSC experiments (*Shaw et al., 2004*), our

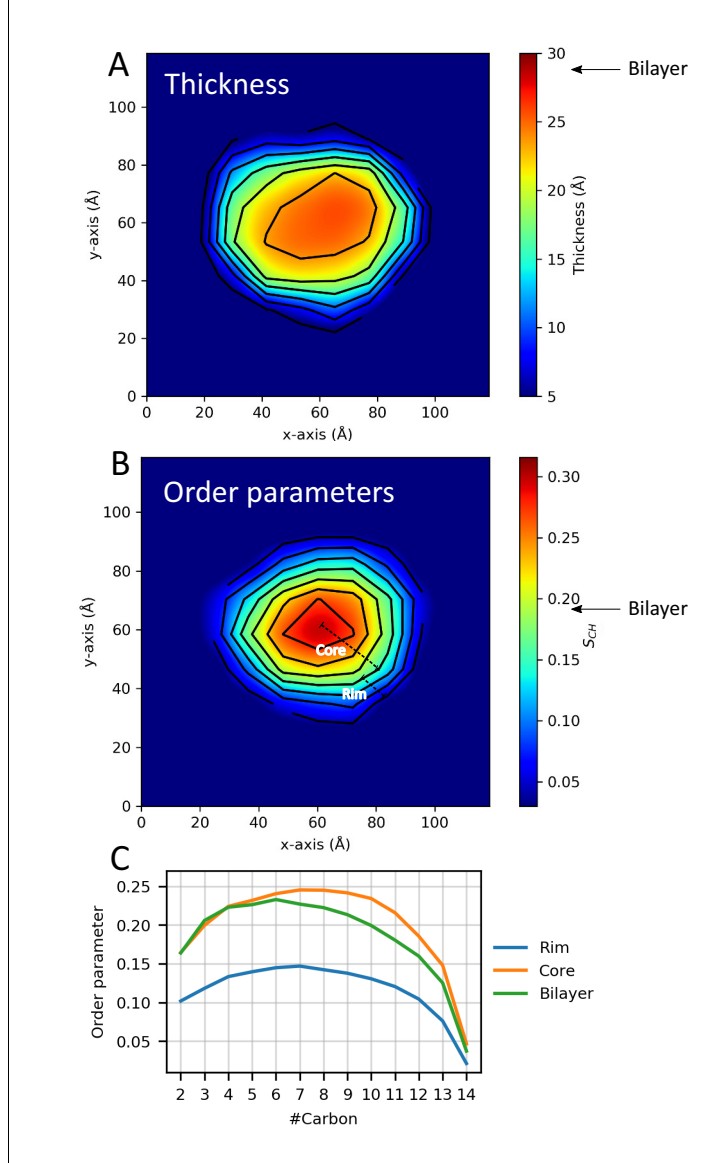

**Figure 4.** Lipid properties from simulations. 2D plots of the (**A**) thickness and (**B**) order parameters averaged over the ensemble and all C-H bonds in the two aliphatic tails of the DMPC lipids in the ΔH5 nanodisc. The core and rim zones are indicated in panel B. Arrows indicate the average value in simulations of a DMPC bilayer. (**C**) The order parameters as a function of carbon number in the lipid tails in the ΔH5-DMPC disc. The rim zone is defined as all lipids within 10 Å of the MSPs, while the core zone is all the lipids not within 10 Å of the MSPs.

results show that the melting transition peak broadens significantly in all three nanodisc systems compared to that of pure DMPC vesicles (*Figure 5*). The broader melting transition is in line with the observed differentiated lipid ordering in nanodiscs from the reweighted simulations, as the observed differences in how ordered the lipids are necessarily will cause differences in the melting temperature and thus give rise to the broader peaks. Furthermore, the broadened peaks are in line with results observed in previous solid state NMR experiments which found a substantially broadened and diminished lipid gel-liquid phase transition in the ΔH5-DMPC nanodisc in the temperature range 10-28°C (*Martinez et al., 2017*). Our results show that the transition enthalpy per mole of DMPC, i.e. the area under the curves, increases with the nanodisc size, in line with previous observations for, respectively, DMPC and DPPC in MSP1D1 and in the larger MSP1E3D1 systems (*Denisov et al., 2005*), where it was proposed to be due to the absence of a cooperative melting transition of the lipids at the nanodisc rim (*Denisov et al., 2005*).

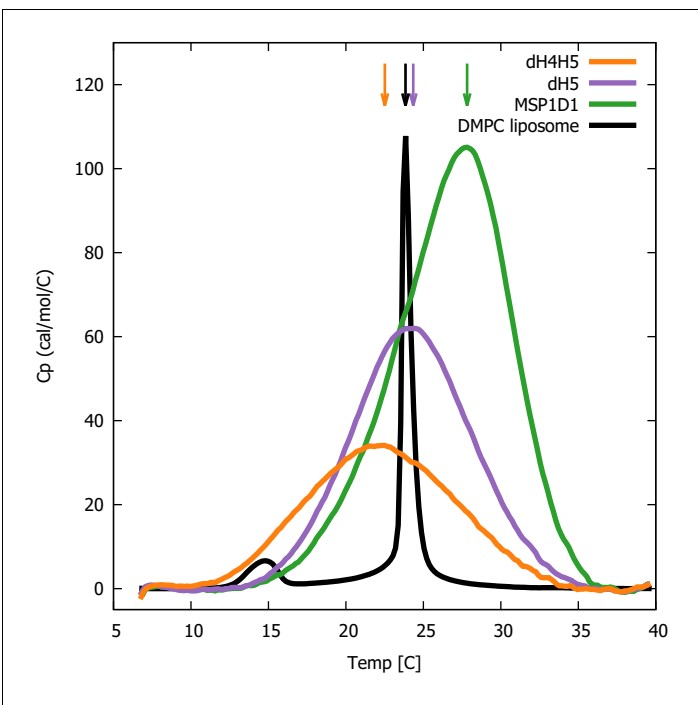

**Figure 5.** DSC analysis of lipid melting in nanodiscs. The three DMPC-filled nanodiscs studied, listed by increasing size, are ΔH4H5-DMPC (orange), ΔH5-DMPC (purple) and MSP1D1-DMPC (green). DSC data from plain DMPC-vesicles (black) are shown for comparison. Arrows indicate the temperature with maximal heat capacity. DSC data from the three nanodisc samples are normalized by DMPC concentration, while the data from the DMPC liposome is on an arbitrary scale.

Interestingly, we observe that the maximum of the melting transition, $T_M$, depends on nanodisc belt and can fall both below and above the $T_M$ of plain DMPC vesicles (24°C). In the smallest ΔH4H5 nanodisc, the DMPC has a $T_M \approx$ 22.5°C. In ΔH5 the DMPC has $T_M$ at 24.5°C which is close to that of the DMPC vesicles, while the larger MSP1D1 nanodisc has a $T_M \approx$ 28°C. This $T_M$ value is similar to the value of 28.5°C measured for DMPC melting in MSP1D1 by *Denisov et al., 2005*, who in addition measured a $T_M$ value of 27.5°C in the even larger MSP1E3D1 discs.

Together our results are in line with previous NMR experiments (*Martinez et al., 2017*), and suggest that the state of the ordering of the lipids in the nanodiscs is inhomogeneous compared to the DMPC vesicle, and that the behaviour of the lipids is modulated by their interaction with the membrane scaffold proteins. Our results point towards a non-trivial effect of the DMPC-MSP interactions. They can both destabilize DMPC in the gel-phase in the smaller nanodiscs (ΔH4H5-DMPC) where the low area-to-rim ratio leads to the lower $T_M$ compared to the DMPC vesicles, but also stabilize the DMPC gel-phase in larger nanodiscs with larger area-to-rim ratios such as MSP1D1-DMPC and MSP1E3D1-DMPC. Thus, when using nanodiscs as membrane mimics it is relevant to keep in mind that the given lipid gel/liquid state might be affected. We also note that even if lipids in larger discs are less perturbed than those in the smallest discs, introduction of membrane proteins into the discs might in itself perturb the lipids in ways similar to the MSPs.

## Conclusions

Lipid nanodiscs are versatile membrane mimetics with a wide potential for studies of the structure, function and dynamics of membrane proteins. Despite their widespread use and numerous studies, we still do not have a full and detailed understanding of the structural and dynamic features of nanodiscs. This in turn limits our ability to interpret e.g. solution scattering experiments when membrane proteins are embedded into such nanodiscs. In order to further our understanding of the

conformations and structural fluctuations of both the protein and lipid components in nanodiscs, we have performed a series of biophysical experiments on DMPC-loaded ΔH5 and ΔH4H5 nanodiscs.

Using SEC-SAXS and SEC-SANS measurements, we investigated the solution structure of the ΔH4H5-DMPC and ΔH5-DMPC discs. Model-based analysis of this data showed an 'average' elliptical shape of both nanodiscs. In contrast, a previously determined integrative NMR/EPR (*Bibow et al., 2017*) method gave rise to a more circular average structure of the ΔH5 nanodisc.

We reconcile these two apparently opposing views and provide a richer and more detailed view of the nanodisc proteins and lipids and their dynamics by performing MD simulations. In particular, we used a Bayesian/Maximum Entropy approach to integrate the MD simulations with the SAXS and NMR data to uncover the existence of underlying fluctuations between elliptical shapes with orthogonal major axes in consistency with both sources of data. We note that the NMR/EPR-derived structure, and our MD simulations initiated from this structure, provide good agreement with the SAXS data even without reweighting. Because our SAXS data are rather precise, however, we were able to detect subtle deviations that enabled us to refine our model. An interesting avenue for further analysis might be to use our structural ensembles to interpret electron microscopy data of nanodiscs. Negative stain transmission electron microscopy of ΔH5 appears to show discs of different shapes (*Bibow et al., 2017*), whereas class-averaged cryo-electron microscopy of a membrane protein embedded in a different nanodisc appears more symmetric (*Frauenfeld et al., 2011*). Direct comparisons between solution structures and electron microscopy data should also take into account any possible changes in shape that might happen during the freezing process. We have previously used contrast-variation to prepare specifically deuterated nanodiscs that become invisible to neutrons in $D_2O$ (*Maric et al., 2014*). In the future it would be interesting to use a similar strategy to study the belt proteins and lipids independently by matching out each component separately.

In addition to studying the overall shape fluctuations, we also analyzed the lipid structure and dynamics in the nanodiscs, and find an inhomogeneous distribution. Specifically, we find substantially perturbed lipid properties near the belt proteins, whereas the lipids more central in the disc behaved more similar to those in a pure DMPC bilayer. We used DSC to investigate the lipid melting transition in the small nanodiscs in comparison to the lipid vesicles and found that the melting takes place over a much broader temperature range in the small nanodiscs. The observed correlation between the size of the belt proteins and the lipid melting enthalpy give support to the proposition (*Denisov et al., 2005*), that the arrangement of the lipids near the nanodisc rim must be substantially perturbed. In particular, our results suggest that the belt proteins induces additional disorder to the lipid tails near the rim.

Together, our results provide an integrated view of both the protein and lipid components of nanodiscs. Approaches such as the one described here takes advantage of the increasing possibilities for accurate NMR and scattering data in solution, improved computational models for lipid bilayers as well as new approaches to integrate experiments and simulations. In this way, our study exemplifies how integrating multiple biophysical experiments and simulations may lead to new insight into a complex system and paves the way for future studies of membrane proteins inside nanodiscs.

## Materials and methods

### Expression of membrane scaffold protein (MSP) variants

We used previously reported constructs for ΔH4H5, ΔH5 and MSP1D1 (*Hagn et al., 2013*; *Ritchie et al., 2009*). We expressed and purified the proteins as previously described (*Ritchie et al., 2009*), with minor modifications to the purification protocol: The cells were opened in lysis buffer containing 50 mM Tris/HCl pH 8.0, 300 mM NaCl, 1% Triton X-100 and 6 M GuHCl by vigorous shaking for 15 min. Insoluble material was subsequently removed by centrifugation at 18,000 rpm for 1 hr using an SS-34 rotor. The supernatant was loaded on Ni-NTA resin pre-equilibrated in lysis buffer and washed extensively with the same buffer. Extensive washes using lysis buffer without GuHCl and subsequently wash buffer containing 50 mM Tris/HCl pH 8.0, 300 mM NaCl, 20 mM Imidazole and 50 mM Cholate was performed in order to remove GuHCl and Triton X-100. Protein was eluded in buffer containing 50 mM Tris/HCl pH 8.0, 300 mM NaCl, 500 mM Imidazole, concentrated, flash frozen and stored at −80°C until further use. Cleavage of the TEV-site was performed by addition of

1:20 TEV protease, and dialysing at room temperature for 6–12 hr against 20 mM TrisHCl pH 8, 100 mM NaCl, 0.5 mM EDTA, 1 mM DTT. TEV protease and any un-cleaved MSP was removed by passing the solution over Ni-NTA resin again.

## Reconstitution of ΔH5-DMPC and ΔH4H5-DMPC nanodiscs

Before assembly, the DMPC lipids (Avanti Polar Lipids) were suspended in a buffer containing 100 mM NaCl, 20 mM Tris/HCl pH 7.5, and 100 mM sodium cholate detergent to a final lipid concentration of 50 mM. We determined optimal reconstitution ratios between the DMPC lipids and the ΔH5 and ΔH4H5 by first mixing the lipid and MSP stock solutions at a series of different molar concentration ratios in the range from 1:9 to 1:80 depending on the MSP type (*Figure 1—figure supplement 1*). In all samples, cholate was removed after mixing by addition of an excess amount of Amberlite detergent absorbing beads to start the assembly of the nanodiscs. The samples were left in a thermomixer for 4 hr at 28°C and the Amberlite was removed by centrifugation at 5000 rpm. Purification was performed using size exclusion chromatography (SEC) on an Äkta purifier (FPLC) system with a Superdex200 10/300 column from (GE Healthcare Life Science; S200). We found that reconstitution ratios of 1:33 for ΔH4H5:DMPC and 1:50 for ΔH5:DMPC resulted in a single and relatively narrow symmetric peak, in good agreement with the previously reported ratios of 1:20 for ΔH4H5:DMPC and 1:50 for ΔH5:DMPC (*Hagn et al., 2013*). More narrow and well-defined SEC-peaks were obtained if the reconstitution took place at or above the melting temperature, $T_M$, of DMPC at 24°C (*Ritchie et al., 2009*).

## Differential scanning calorimetry (DSC)

The measurements were performed on a VP-DSC (MicroCal) using a constant pressure of 1.7 bar (25 psi) and a scan rate of 1°C/min between 6°C and 40°C. All samples had been purified in PBS buffer prior to the measurement. We used the Origin instrument software for background subtraction and baseline correction using a 'Cubic Connect' baseline correction. Finally, the data were normalized by the lipid concentration of the individual samples.

## SEC-SANS

SEC-SANS was performed at the D22 small-angle scattering diffractometer at the ILL, Grenoble, France using a recently developed SEC-SANS setup (*Jordan et al., 2016*; *Johansen et al., 2018*). Briefly, the setup was as follows: the in situ SEC was done using a modular HPLC system (Serlabo) equipped with a Superdex 200 GL gel filtration column (GE) with a void volume of approximately 7.5 ml and a flow rate of 0.25 ml/min. The SmartLine 2600 diode-array spectrophotometer (Knauer) was connected via optic fibers either to an optic cell of 3 mm path length placed at the outlet of the chromatography column, enabling the simultaneous recording of chromatograms at four different wavelengths, including 280 nm which we used for the concentration determination. All components of the HPLC setup including buffers and the column were placed in a closed cabinet connected to an air-cooling system set to 10°C to control the temperature of the sample. Before measurements, we equilibrated the column in a $D_2O$-based buffer, and the buffer in the sample was exchanged to a $D_2O$-based buffer using an illustra NAP-25 gravity flow column (GE). The $D_2O$ buffer contained 20 mM Tris/DCl pH 7.5 and 100 mM NaCl.

The experiments were carried out with a nominal neutron wavelength, λ, of 6.0 Å and a wavelength distribution, $\Delta\lambda/\lambda = 10\%$ FWHM, a rectangular collimation of 40 mm × 55 mm and a rectangular sample aperture of 7 mm × 10 mm. The distance of the sample-detector used for the characterization of the nanodiscs was 5.6 m (with collimation of 5.6 m), covering a momentum transfer range, q, of 0.0087 Å$^{-1}$ to 0.17 Å$^{-1}$, with $q = 4\pi\sin(\theta)/\lambda$, where θ is half the angle between the incoming and the scattered neutrons. Measured intensities were binned into 30 s frames. Sample transmission was approximated by the buffer measured at the sample-detector distance of 11.2 m. The measured intensity was brought to absolute scale in units of scattering cross section per unit volume (cm$^{-1}$) using direct beam flux measured for each collimation prior to the experiment. Data reduction was performed using the GRASP software (https://www.ill.eu/fr/users-en/scientific-groups/large-scale-structures/grasp/). The SANS data appropriate for buffer subtraction was identified based on when the 280 nm absorption during the SEC curve showed no trace of protein.

## SEC-SAXS

SEC-SAXS was performed at the BioSAXS instrument at BM29 at the ESRF, Grenoble, France (*Pernot et al., 2013*). Briefly, the setup at BM29 included an HPLC controlled separately from the SAXS measurement, coupled to a UV-Vis array spectrophotometer collecting absorption from 190 nm to 800 nm. Data were collected with a wavelength of 0.9919 Å using a sample-detector distance of 2.87 m which provided scattering momentum transfers ranging from 0.003 Å$^{-1}$ to 0.49 Å$^{-1}$. The capillary was cooled to 10℃, however, the HPLC including the SEC-column was placed at ambient temperature. Size exclusion chromatography was performed using the same column as for SEC-SANS and equivalent H$_2$O-based buffer. A flow rate of 0.5 ml/min was used. Data reduction was carried out using the in-house software, and subsequent conversion to absolute units was done with water as calibration standard (*Orthaber et al., 2000*). The 1 s frames recorded were subsequently averaged in 10 s bins.

## Standard solution SAXS

Standard solution SAXS data were obtained at the P12 beamline at the PETRA III storage ring in Hamburg, Germany (*Blanchet et al., 2015*) using a wavelength of 1.24 Å, a sample-detector distance of 3 m, providing a momentum transfers covering from 0.0026 Å$^{-1}$ to 0.498 Å$^{-1}$ and a variable temperature in the exposure unit. 20 exposures of 0.045 s were averaged, background subtracted and normalized to absolute scale units (cm$^{-1}$) using Bovine Serum Albumin, BSA as calibration standard by the available software at the beamline. The measurements were performed at both 10℃ and 30℃.

## SAXS and SANS data analysis

The output of the SAXS and SANS experiments were small-angle scattering data in terms of absolute intensities $I(q)$. $I(q)$ was transformed into the pair distance distribution function, $p(r)$, by indirect Fourier transformations using BayesApp (*Hansen, 2014*). Further SAXS/SANS modelling was carried out using our previously developed WillItFit software (*Pedersen et al., 2013*) (https://sourceforge. net/projects/willitfit/). The applied structural models (see further description below) are an adaptation of similar models previously developed to analyse SAXS and SANS data from MSP1D1 nanodiscs (*Skar-Gislinge et al., 2010*; *Skar-Gislinge and Arleth, 2011*). Briefly, the model describes the nanodiscs as coarse-grained elliptical shapes and is based on analytical form factors (*Pedersen, 1997*; *Skar-Gislinge et al., 2010*; *Skar-Gislinge and Arleth, 2011*). The ellipticity, in terms of the axis ratio of the embedded bilayer patch is allowed to vary in the fit and can also take the size of unity corresponding to a circular disc. The model is fitted on absolute scale and utilizes information on the composition of the protein belt and lipids, and the molecular volumes, $v$, of the DMPC lipids and the different belts with/without His-tag. These are taken to be $v_{DMPC} = 1085\text{Å}^{-3}$, $v_{\Delta H4H5} = 20349\text{Å}^{-3}$, $v_{\Delta H4H5} = 24298\text{Å}^{-3}$, $v_{His} = 3142\text{Å}^{-3}$. The X-ray and neutron scattering lengths of the different components are calculated from their chemical composition.

Apart from the parameters listed in *Table 1*, the model also fits a small constant background added to the model, and includes a term accounting for interface roughness, fixed to 2 Å in the present analysis, and where relevant, a Gaussian random coil description of the linked TEV-His-tag with $R_G$ = 12.7 Å consistent with the assumption that the 23 amino acids of the tag are in a fully disordered state (*Kohn et al., 2004*). As our measurements are on a calibrated absolute intensity scale, we can compare the observed intensities with those expected from the composition of the sample. Both the SAXS and SANS data had to be re-scaled by a constant close to unity to fit the data (*Table 1*), but in the case of the ΔH4H5-DMPC SANS data, the scaling constant (1.7 ± 0.5) was larger than expected, most likely the result of a less accurate protein concentration determination for this system.

## MD simulations

We initiated our MD simulations from the first model in PDB ID 2N5E (*Bibow et al., 2017*). A total of 50 pre-equilibrated DMPC lipids (*Domański et al., 2010*; *Dickson et al., 2012*) were inserted into each monolayer inside the protein belt. The number of lipids was chosen from the measured optimal reconstitution ratio, and in accordance with the reconstitution ratio used in the experiments for the NMR structure (*Bibow et al., 2017*) as well as obtained from our fit of the geometric model to the

SAXS and SANS data. The MD simulations were performed using GROMACS 5.0.7 (*Pronk et al.,* *2013*; *Abraham et al., 2015*) and the CHARMM36m force field (*Huang et al., 2017*). The system was solvated in a cubic box and neutralized by addition of Na$^+$ counter ions followed by a minimization of the solvent. Equilibration was performed in six steps following the protocol from CHARMM-GUI (*Lee et al., 2016*) with slow decrease in the positional restraint forces on both lipids and protein. The volume of the box was then equilibrated in the NPT ensemble at 303.15 K and 1 bar giving a final box with side lengths 13.2 nm. The production run was performed in the NVT ensemble at 303.15 K (above the phase transition of the DMPC lipids) using the stochastic velocity rescaling thermostat (*Bussi et al., 2007*), 2 fs time steps and the LINCS algorithm to constrain bonds. We performed two production runs (lengths 600 ns and 595 ns) starting from the same equilibrated structure. We concatenated these two MD simulations into a single trajectory, which then represents our sample of the dynamics of the system. We clustered the conformations from the simulations (one structure extracted for every nanosecond) with the RMSD based Quality Threshold method (*Heyer et al., 1999*; *Melvin et al., 2016*) using C$_\alpha$ atoms only and with a cluster diameter cutoff of 0.58 nm; this resulted in six clusters. We also performed a 50ns-long simulation of a pure DMPC bilayer. The simulation parameters were the same as for the nanodisc system apart from using the NPT ensemble and anisotropic pressure control.

## Calculating SAXS and SANS from simulations

We performed SAXS calculations using both CRYSOL (*Svergun et al., 1995*) and FOXS (*Schneid-man-Duhovny et al., 2013*; *Schneidman-Duhovny et al., 2016*) on structures extracted every 1 ns from the simulations and for the $q$-range from 0.0 Å$^{-1}$ to 0.25 Å$^{-1}$. Most of the overall structural information is contained within this $q$-range, and the calculations of SAXS intensities from the structures are also less accurate in the wide-angle regime. We used standard solution SAXS data recorded at 30˚C on the ΔH5-DMPC (without His-TEV-tags) to compare to our simulations, as this setup is most similar to that used to derive the NMR/EPR structure. The SAXS profile of the NMR/EPR structure was calculated by adding DMPC lipids to the first model of the PDB entry and subsequent equilibration of the lipids by MD (fixing the protein), and then using FOXS to back-calculate the SAXS.

Both CRYSOL and FOXS are implicit solvent methods that use fitting parameters to take into account the buffer subtraction and the solvation layer around the solute. The programs automatically optimize these parameters by fitting to experimental data for each input frame, but applying this approach to many frames in a molecular dynamics trajectory could lead to over-fitting. Instead, we calculated the average of each fitted parameter over the trajectory and re-calculate the SAXS with the parameters fixed to this average. FOXS has two parameters, $c1$ (scaling of atomic radius for adjustment of excluded volume) and $c2$ (solvation layer adjustment) which, after the fitting, are set to small intervals around the averages $[1.01 : 1.02]$ and $[-0.148 : -0.140]$, respectively. Narrow intervals are used as the program only takes an interval for the parameters. CRYSOL's fitting parameters *dro* (Optimal hydration shell contrast), *Ra* (Optimal atomic group radius) and *ExVol* (relative background) are set to $[0.0090 : 0.0098]$, $[1.72 : 1.76]$ and $[162300 : 162320]$, respectively. Both CRYSOL and FOXS calculations were performed with hydrogens explicitly included in order to limit artifacts from the excluded volume parameter settings, that is, buffer subtraction, that is suspected to arise from the lipid tails (*Chen and Hub, 2015*). For CRYSOL the additional settings Maximum order of harmonics was set to 50, the Order of Fibonacci grid to 18 while the Electron density of the solvent was set to 0.334 $e$/Å$^3$ .

SANS calculations were performed using CRYSON (*Svergun et al., 1998*) setting the maximum order of harmonics to 50, the order of the Fibonacci grid to 18 and the fraction of D$_2$O in solution to 1.0 in accordance with the experimental measurements. The experimental SANS data were measured on a His-TEV tagged nanodisc. For comparison, we used the simulation frames and added His-TEV tags computationally by extracting conformations from our simulation (w/o His-tags) every 1 ns and attaching a random His-tag structure generated from Flexible Meccano (*Ozenne et al.,* *2012*) and Pulchra (*Rotkiewicz and Skolnick, 2008*) from a pool of 10000 structures to the tails of the nanodisc. If there we detected any clash of the attached His-TEV-tag structure with the protein belt or lipids of the nanodisc or with the second His-TEV-tag, the His-TEV-tag was discarded and a new random structure from the pool was attached. By sampling randomly from a pool of 10.000 His-tag structures together with having in total 1195 frames from the simulation of the nanodisc (1ns per

frame) we assume that the His-TEV-tags represents a sufficiently realistic distribution to model the impact on the SANS data.

## Comparing simulations to NOEs

We calculated distances corresponding to the experimentally observed NOEs on structures extracted every 1 ns from the simulations. To compare with the experimental distances, available as upper bounds, we averaged the distances, $R$, between the respective atoms (or the geometric center for pseudo atoms) as $<R^{-3}>^{-1/3}$ (*Tropp, 1980*). When calculating $\chi^2$ for validation we only include those distances where this average exceeded the experimentally-determined upper-bounds.

## Calculating EPR and PRE data from simulations

We used a previously developed rotamer library for MTSL spin-label probes (*Polyhach et al., 2011*; *Klose et al., 2012*) to calculate both EPR and PRE data using the DEER-PREdict software (*Tesei et al., 2020*; https://github.com/KULL-Centre/DEERpredict). In the case of the EPR DEER data, we calculated the distance distribution of spin-label probes and compared to those estimated from experiments (*Bibow et al., 2017*). For the NMR data we used a Model Free approach to calculate the PREs (*Iwahara et al., 2004*) and estimated intensity ratios as previously described (*Battiste and Wagner, 2000*) using $R_{2,dia} = 60 \text{s}^{-1}$, $\tau_c = 34 \text{ns}$, $\tau_t = 1 \text{ns}$ and an INEPT delay of 10 ms.

## Integrating experiments and simulations

We used a Bayesian/maximum entropy approach (*Różycki et al., 2011*; *Hummer and Köfinger, 2015*; *Bottaro et al., 2018*), as implemented in the BME software (*Bottaro et al., 2020*) (github. com/KULL-Centre/BME), to integrate the molecular simulations with the SAXS and NMR experiments. The name originates from the two equivalent approaches, Bayesian and Maximum Entropy ensemble refinement, which are equivalent when the errors are modelled as Gaussians (*Hummer and Köfinger, 2015*; *Cesari et al., 2016*; *Bottaro et al., 2020*). We here provide a brief overview of the approach and refer the reader to recent papers for more details (*Hummer and Köfinger, 2015*; *Cesari et al., 2016*; *Bottaro et al., 2020*; *Orioli et al., 2020*).

Given that our MD simulations provide a good, but non-perfect, agreement with experiments the goal is to find an improved description of the nanodisc that simultaneously satisfies two criteria: (i) the new ensemble should match the data better than the original MD ensemble and (ii) the new ensemble should be a minimal perturbation of that obtained in our simulations with the CHARMM36m force field in accordance with the maximum entropy principle. In a Bayesian formulation, the MD simulation is treated as a *prior* distribution and we seek a *posterior* that improves agreement with experiments. This may be achieved by changing the weight, $w_j$, of each conformation in the MD-derived ensemble by minimizing the negative log-likelihood function (*Hummer and Köfinger, 2015*; *Bottaro et al., 2020*):

$$\mathcal{L}(w_1 \ldots w_n) = \frac{m}{2} \chi_r^2(w_1 \ldots w_n) - \theta S_{rel}(w_1 \ldots w_n). \tag{1}$$

Here, the reduced $\chi_r^2$ quantifies the agreement between the experimental data ($F_i^{EXP}$) and the corresponding ensemble values, ($F(\mathbf{x})$), calculated from the weighted conformers ($\mathbf{x}$):

$$\chi_r^2(w_1 \ldots w_n) = \frac{1}{m} \sum_i^m \frac{(\sum_j^n w_j F_i(\mathbf{x}_j) - F_i^{EXP})^2}{\sigma_i^2}. \tag{2}$$

The second term contains the relative entropy, $S_{rel}$, which measures the deviation between the original ensemble (with initial weights $w_j^0$ that are uniform in the case of a standard MD simulation) and the reweighted ensemble $S_{rel} = -\sum_j^n w_j \log\left(\frac{w_j}{w_j^0}\right)$. The temperature-like parameter $\theta$ tunes the balance between fitting the data accurately (low $\chi_r^2$) and not deviating too much from the prior (low $S_{rel}$). It is a hyperparameter that needs to be determined (*Figure 3—figure supplement 3*). In practice it turns out that minimizing $\mathcal{L}$ can be done efficiently by finding Lagrange multipliers in an equivalent Maximum Entropy formalism and we refer the reader to previous papers for a full description and discussion of the approaches including how to determine $\theta$ (*Hummer and Köfinger, 2015*;

*Cesari et al., 2016*; *Bottaro et al., 2020*). The weights from the BME analysis, the MD simulations as well as the various data that we analyzed are available online at https://github.com/KULL-Centre/papers/tree/master/2020/nanodisc-bengtsen-et-al (copy archived at https://github.com/elifesciences-publications/KULL-CENTREpapers/tree/master/2020/nanodisc-bengtsen-et-al; *KULL-Centre, 2020*).

## Acylindricity

In order to quantify how 'elliptical' the different nanodisc conformations are, we calculated the square root of the acylindricity, $\sqrt{C}$, where the acylindricity is defined from the principal components of the gyration tensor as $C := \lambda_x^2 - \lambda_y^2$, where the z-axis is orthogonal to the membrane and has the smallest principal component. In our calculations we included only the protein backbone atoms (excluding also the flexible tails from residues 55–63). This choice also makes it possible to compare with a similar calculation from the geometric model fitted from the SAXS and SANS data where the acylindricity was calculated using the major and minor axes from the geometric fit.

## Lipid properties

We calculated the bilayer thickness and lipid order parameters for both the nanodisc and a simulated DMPC lipid bilayer. The values obtained for the nanodisc were from the reweighted ensemble every 1 ns. We defined the bilayer thickness as the minimum distance along the bilayer normal between two phosphate headgroup pairs in the two leaflets. The headgroup pairs were identified and saved for each leaflet, top and bottom, along with the corresponding thickness and xy-coordinates. The pairs were further distributed unto a $6 \times 6$ grid in the xy-plane with each bin corresponding to 22 Å for both the top and bottom leaflet. An averaged grid was then obtained from the two grids of the leaflets. The order parameters $S_{CH}$ where calculated as (*Piggot et al., 2017*): $S_{CH} = \frac{1}{2}\langle 3cos^2\theta - 1 \rangle$, where θ is the angle between the C-H bond and the bilayer normal. The order parameters were calculated for each lipid and each carbon along the two lipid tails every 1 ns. The values were further averaged across the two lipid tails before distributed unto a $6 \times 6$ grid. An average across frames and lipids were then obtained for each bin. In order to study the profile of the lipid tails, an average across frames, lipids, and tails were likewise obtained. Parameters were calculated from the simulations of the DMPC bilayer in the same way.

## Acknowledgements

We thank Stefan Bibow, Gunnar Jeschke, Yevhen Polyhach and Roland Riek for discussions and input, and for sharing experimental data, and Jesper Ferkinghoff-Borg for discussions and input to analysis of lipid structures. The research described here was supported by a grant from the Lundbeck Foundation to the BRAINSTRUC structural biology initiative (LA and KL-L), a Hallas-Møller Stipend from the Novo Nordisk Foundation (KL-L), the Velux Foundations (KL-L) and the Danish Council for Independent Research in Technology and Production (BS).

## Additional information

### Funding

| Funder | Grant reference number | Author |
| --- | --- | --- |
| Danish Council for Independent Research | | Birgit Schiøtt |
| Lundbeckfonden | BRAINSTRUC | Lise Arleth Kresten Lindorff-Larsen |
| Novo Nordisk Foundation | Hallas-Møller Stipend | Kresten Lindorff-Larsen |
| Villum Fonden | Block grant | Sandro Bottaro Kresten Lindorff-Larsen |

The funders had no role in study design, data collection and interpretation, or the decision to submit the work for publication.

## Author contributions

Tone Bengtsen, Data curation, Software, Formal analysis, Investigation, Visualization, Methodology, Writing - original draft, Writing - review and editing; Viktor L Holm, Data curation, Formal analysis, Investigation, Writing - original draft; Lisbeth Ravnkilde Kjølbye, Software, Formal analysis, Visualization, Methodology, Writing - review and editing; Søren R Midtgaard, Formal analysis, Investigation, Writing - review and editing; Nicolai Tidemand Johansen, Data curation, Formal analysis, Investigation; Giulio Tesei, Sandro Bottaro, Software, Formal analysis, Methodology; Birgit Schiøtt, Resources, Supervision, Methodology, Writing - review and editing; Lise Arleth, Conceptualization, Resources, Formal analysis, Supervision, Funding acquisition, Methodology, Project administration, Writing - review and editing; Kresten Lindorff-Larsen, Conceptualization, Resources, Formal analysis, Supervision, Funding acquisition, Investigation, Methodology, Project administration, Writing - review and editing

## Author ORCIDs

Giulio Tesei (iD) https://orcid.org/0000-0003-4339-4460
Sandro Bottaro (iD) http://orcid.org/0000-0003-1606-890X
Kresten Lindorff-Larsen (iD) https://orcid.org/0000-0002-4750-6039

## Decision letter and Author response

Decision letter https://doi.org/10.7554/eLife.56518.sa1
Author response https://doi.org/10.7554/eLife.56518.sa2

# Additional files

## Supplementary files

• Transparent reporting form

## Data availability

Scattering data, molecular simulations and results from reweighting are available at https://github.com/KULL-Centre/papers/tree/master/2020/nanodisc-bengtsen-et-al (copy archived at https://github.com/elifesciences-publications/KULL-CENTREpapers/tree/master/2020/nanodisc-bengtsen-et-al).

The following previously published dataset was used:

| Author(s) | Year | Dataset title | Dataset URL | Database and Identifier |
|---|---|---|---|---|
| Bibow S, Polyhach Y, Eichmann C, Chi CN, Kowal J, Stahlberg H, Jeschke G, Guentert P, Riek R | 2017 | The 3D solution structure of discoidal high-density lipoprotein particles | http://www.rcsb.org/structure/2N5E | RCSB Protein Data Bank, 2N5E |

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
