## [Decision Letter]

**Acceptance summary:**

Computer simulations are increasingly used in structural biology as a means to integrate different kinds of experimental biophysical data, so as to facilitate a coherent and physically realistic molecular-level interpretation. This paper reports a structural refinement strategy of a previously determined NMR structure of a lipid nanodisc using orthogonal scattering data and advanced MD simulations. The elegant integrative simulation/experimental Ansatz leads to the insightful conclusion that the nanodisc undergoes elliptical structural fluctuations. Such elliptic fluctuations are "averaged out" in the NMR signal, but not in the scattering experiments data, thus resolving the apparent discrepancy between the different experimental methods.

**Decision letter after peer review:**

Thank you for submitting your article "Structure and dynamics of a nanodisc by integrating NMR, SAXS and SANS experiments with molecular dynamics simulations" for consideration by *eLife*. Your article has been reviewed by four peer reviewers, including Lucie Delemotte as the Reviewing Editor and Reviewer #1, and the evaluation has been overseen by José Faraldo-Gómez as the Senior Editor. The following individual involved in review of your submission has agreed to reveal their identity: Franz Hagn (Reviewer #3).

The reviewers have discussed the reviews with one another and the Reviewing Editor has drafted this decision to help you prepare a revised submission.

Summary:

The present manuscript builds up on previous work by the same authors (e.g. Skar-Gislinge, 2010, Midtgaard, 2015) towards the goal of understanding the structure and dynamics of MSP1D1-ΔH5 and MSP1D1-ΔH4H5 nanodiscs and their membrane mimicking capabilities. In particular, small angle X-ray and neutron scattering experiments are performed and used together with previously determined NMR distance restraints (Bibow et al., 2017)) and molecular dynamics simulations to obtain additional information on the structural properties of a lipid nanodisc particle in solution.

In essence, this paper reports on a structural refinement strategy of a previously determined NMR structure of a lipid nanodisc using orthogonal scattering data and advanced MD simulations. According to a procedure established by the Lindorff-Larsen lab, the conformational clusters obtained in the simulations are re-weighted to obtain a better fit to the experimental data.

A key question addressed by the authors is whether the NDs are circular, as suggested by the previous NMR experiments, or slightly elliptical, as suggested by the scattering experiments carried out by the authors. The elegant integrative simulation/experimental Ansatz leads to the insightful conclusion that the ND undergoes elliptical structural fluctuations, with the major axes changing direction and being largely orthogonal to each other. Such elliptic fluctuations are "averaged out" in the NOEs from NMR, but not in the SAXS data, thus resolving the apparent discrepancy between the different experimental methods.

Essential revisions:

1) The reviewers agreed unanimously that the work was methodologically rigorous, technically well done, and the scientific conclusions fully supported by the data. They also agreed that this manuscript was clearly written, and that the presentation of the data was thorough and well organized. However, it was not entirely clear what aspects of this manuscript were novel. SAXS and SANS on MSP1D1 nanodiscs, lipid melting transition temperature dependence probed by SAXS on MSP1D1 and MSP1E3D1, analysis of lipid properties in MSP1 nanodiscs, broadening of gel-liquid phase transition in MSP1D1-ΔH5, size-dependent increase of the transition enthalpy in MSP1D1 and MSP1E3D1 had been reported beforehand. The MD methodology as well as SAXS/SANS model building and fitting has been previously published by the authors.

It thus appeared to the reviewers that the novel aspect of this paper was the combination of various structural data and simulations to obtain better structures of challenging systems. Thus, while this paper provided a valuable protocol for dealing with structural data of various sources, this was not entirely clear from the paper in the present form.

Additionally the relevance of an oval nanodisc shape for their membrane mimicking properties was not entirely clear, since the shape might change in presence of a membrane protein. It was also unclear whether the conclusions around ellipticity could be generalized for all nanodiscs.

The reviewers thus recommended that the paper be rewritten to reflect the novel aspects of the work and to highlight its strengths while clarifying which aspects were already tackled by previous work.

2) In addition to the circular NMR structure from Bibow et al., 2017, a circular nanodisc structure was solved in a cryo-electron microscopy study by Frauenfeld et al., 2011. How would the authors compare their results to this single particle cryo-electron microscopy structure? It would be interesting to use also these restraints into the MD simulation. Could the oval shape of an empty nanodisc also be seen in (cryo-)electron microscopy single particle data (before class averaging)?

3) The work by Bibow et al. also includes PRE distance restraints and EPR distance restraints (and for the latter also a size distribution) which are of medium range and thus on almost a similar level of information as the SAXS and SANS data. These restraints should also be used in the reweighing procedure and discussed.

4) The authors show the good fit between SAXS and SANS data towards an elliptical model. The authors should illustrate how a circular model would fit with the data as well as different elliptical models to access the request for an elliptical model.

5) The major problem of any bulk studies is the potential presence of a heterogenous sample likely to be the case for nanodiscs because they may contain a distinct number of lipids. While the authors state that a single SEC peak is present that of course is not sufficient evidence for a single entity. First, this potential issue must be stated. Second, it is suggested to calculate the SAXS and SANS curves for a heterogenous sample with several cyclic nanodiscs having a distinct radius. Third, some argumentation may be put forward that indicate that the heterogeneity is of dynamic nature (such as a single set of NMR peaks, the dynamics of the MD simulation).

6) What do the authors mean with "membrane mimicking capabilities"? Just physical properties of lipids in nanodiscs as compared to liposomes?

7) It is not clearly described how SAXS and SANS data have been scaled for data fitting. Is it just a hardware-dependent correction or does it have an underlying physical meaning? This requires a short explanation.

8) For comparing the NMR structure with the simulation results, have the authors considered the entire NMR structural bundle (10 structures) or just a single structure? This might have a marked impact on the chi-square value for the NMR structure.

9) Correlation of MD with amide NOES (subsection “Integrating experiments and simulations”): The authors might want to add here that the amide-amide NOEs report on the α-helical secondary structure of the individual MSPs and are not quite sensitive to slight changes in the shape of the overall MSP belt. Thus, a good correlation with these restraints is very much expected.

10) Subsection “Integrating experiments and simulations”, seventh paragraph: it seems like the square root(C) value for the NMR/EPR structure is missing.

11) Order parameters (Figure 4B): What C-H moieties in DMPC have been used to calculate the numbers plotted in the figure? Also, it seems counter-intuitive that lipids in the rim region are more mobile than lipids in the core.

12) It is very good that the authors make their MD data publicly available on GitHub. It would be nice if the refined ensemble (or some number of representative frames) could be made available to the scientific community also via a database, such as PDB-dev.

13) The clustering step of the MD simulations seems quite important for the rest of the paper, and the choice of methods and parameters is not rationalized. Were other methods/parameters tested and why did the authors settle on this specific choice?

14) Additionally, the reviewers had the following suggestion, but did not strictly require additional experiments to be performed: Have the authors tried to perform SANS contrast matching experiments to obtain scattering data of the protein belt and the lipid patch only, respectively? Since the NMR structure does not contain lipids, such experiments would be helpful to validate the "lipid-free" structure.

---

## [Author Response]

Essential revisions:1) The reviewers agreed unanimously that the work was methodologically rigorous, technically well done, and the scientific conclusions fully supported by the data. They also agreed that this manuscript was clearly written, and that the presentation of the data was thorough and well organized. However, it was not entirely clear what aspects of this manuscript were novel. SAXS and SANS on MSP1D1 nanodiscs, lipid melting transition temperature dependence probed by SAXS on MSP1D1 and MSP1E3D1, analysis of lipid properties in MSP1 nanodiscs, broadening of gel-liquid phase transition in MSP1D1-ΔH5, size-dependent increase of the transition enthalpy in MSP1D1 and MSP1E3D1 had been reported beforehand. The MD methodology as well as SAXS/SANS model building and fitting has been previously published by the authors.It thus appeared to the reviewers that the novel aspect of this paper was the combination of various structural data and simulations to obtain better structures of challenging systems. Thus, while this paper provided a valuable protocol for dealing with structural data of various sources, this was not entirely clear from the paper in the present form.Additionally the relevance of an oval nanodisc shape for their membrane mimicking properties was not entirely clear, since the shape might change in presence of a membrane protein. It was also unclear whether the conclusions around ellipticity could be generalized for all nanodiscs.The reviewers thus recommended that the paper be rewritten to reflect the novel aspects of the work and to highlight its strengths while clarifying which aspects were already tackled by previous work.

We have shortened and revised the manuscript in an attempt to clarify these matters, focusing both on what is new, and how the information might be useful in a wide context:

We and others have previously examined nanodiscs in solution using SAXS and SANS measurements, and used these to obtain information about the overall shape. Given the low resolution of small-angle scattering experiments, and the complexity of the system, this is clearly an underdetermined problem. Our earlier work thus made two simplifying assumptions that we go beyond in the current work. In particular, in our earlier work we (i) fitted the scattering data using a coarse-grained/continuum models of nanodiscs, and (ii) optimized the parameters in this to find a single “average” representation that explained the data.

In the current work we went beyond these assumptions, by (i) adding additional experimental information from (previously published) NMR data, (ii) using all-atom MD simulations to increase the resolution and making it possible to link the structure/shape to both SAXS/SANS and NMR and (iii) used our Bayesian/Maximum Entropy approach to integrate experiments and simulations. Together, this made it possible to provide a “dynamic ensemble” of this MSPΔH5 nanodisc in solution.

As the previous NMR experiments were performed on DMPC-loaded MSPΔH5 nanodiscs, we measured new SAXS/SANS data on the same system. We also measured new scattering data on MSPΔH4ΔH5 discs, as well as calorimetry data on MSPD1 to examine effects of changing the size.

It is correct that similar observations about the melting transitions have been made before, and we have attempted to make this clearer. We, however, use our experimentally-derived structural model to interpret these observations. In this way, we aim to bridge our computational/experimental view of the protein-lipid system to the thermodynamic properties of the discs. As our work is mostly focused on the MSPΔH5 discs, this is the system for which we can make the strongest conclusions. We, however, speculate that some of the observations will also generalize to other (larger) discs, appropriately considering the changed ratio of “rim” vs. “internal” lipids in discs of different sizes. We have integrated the sections discussing lipid properties of the MSPΔH5 discs with our DSC results to make the relationships clearer.

Finally, it is correct that the shape and properties may indeed be affected by embedding a membrane protein. Indeed, this was one of the motivations for this work – to provide a baseline and overall approach for interpreting experimental data on even more complex systems. While the current work goes some way towards this goal, it remains a complex problem.

2) In addition to the circular NMR structure from Bibow et al., 2017, a circular nanodisc structure was solved in a cryo-electron microscopy study by Frauenfeld et al., 2011. How would the authors compare their results to this single particle cryo-electron microscopy structure? It would be interesting to use also these restraints into the MD simulation. Could the oval shape of an empty nanodisc also be seen in (cryo-)electron microscopy single particle data (before class averaging)?

High-resolution, single-particle cryo-electron microscopy data would indeed be very useful as an independent source of information about the shape.

We cannot compare our simulations directly to the refined structure from Frauenfeld as it (i) is a different disc where both the nanodisc belt and the lipids differ from those of our study (ii) contains a large transmembrane protein region and (iii) is the result of substantial averaging over individual particles. Also, depending on how well-defined the location of the membrane protein is in the disc, class-averaging could lead to averaging over multiple quite differently shaped discs, as particle classification is likely driven by the more asymmetric membrane protein.

Bibow et al. presented negative-stain transmission electron microscopy images of their MSPΔH5 discs. While these are of low resolution, they do appear to show different shapes including some that appear substantially elliptical. However, because they are negative-stain images, we have decided not to compare to them directly. Instead, it would be interesting in the future to compare simulations to individual particles in higher-resolution electron microscopy data.

We note, however, that caution should be exerted to control for possible changes during the cooling process, since the dynamical fluctuations and changes in lipid properties might occur on faster time scales, making it unclear to which extent images under cryogenic conditions would reflect properties in solution.

These issues are now discussed briefly in the revised manuscript.

3) The work by Bibow et al. also includes PRE distance restraints and EPR distance restraints (and for the latter also a size distribution) which are of medium range and thus on almost a similar level of information as the SAXS and SANS data. These restraints should also be used in the reweighing procedure and discussed.

We have now compared our simulation ensembles to the PRE and EPR/DEER spin-label data resulting in two new figures (Figure 2—figure supplements 5 and 6). We opted to use the data for validation rather than refinement. Thus, we used a spin-label rotamer library to calculate both the PREs and the distance-distribution from both the unweighted MD simulations, and the ensemble that we reweighted against the scattering and NMR data. We also compared the results to those calculated from the published structure, which used this data as part of the refinement procedure. Overall, we find that (i) the MD simulations are in good agreement with the PRE and EPR data, (ii) the agreement improves after refining the ensemble against the NOE and SAXS data and (iii) the agreement is comparable to that of the structures refined using the PRE and EPR (and other) data.

4) The authors show the good fit between SAXS and SANS data towards an elliptical model. The authors should illustrate how a circular model would fit with the data as well as different elliptical models to access the request for an elliptical model.

We explore this question in two ways.

First, as described in the manuscript, we compare both our (heterogenous) MD simulations and the (more “round”) NMR structure to the SAXS data and show that neither are in perfect agreement with the scattering data. When refining the MD ensemble, which displays a distribution of elliptical shapes, against the SAXS data we increase the population of less-extremely elliptical shapes, so that the final distribution is covers the range between the two “static” structures from NMR and the coarse-grained/continuum model.

Second, we have repeated the fitting of the SAXS data of the ΔH5-ΔHis sample at 30°C using the (static) coarse-grained/continuum model scanning a range of ellipticities (axis ratios 1.0 – 1.6 of the embedded bilayer patch). In this analysis we take into account that the model parameters are correlated, so that the changed axis ratios in part can be compensated by variation in some of the remaining model parameters. The results suggest strongly that a better fit is obtained when the axis-ratios are distinctly different from one. As seen from the new Figure 1—figure supplement 3 the best fits are obtained for an axis-ratio close to 1.3 as reported in Table 1 of the manuscript. For lower values, the model has too pronounced oscillations around q=0.1 Å^-1^ – 0.2 Å^-1^, while for larger values, the calculated values become too “smeared out” and compared to the experimentally observed features.

5) The major problem of any bulk studies is the potential presence of a heterogenous sample likely to be the case for nanodiscs because they may contain a distinct number of lipids. While the authors state that a single SEC peak is present that of course is not sufficient evidence for a single entity. First, this potential issue must be stated. Second, it is suggested to calculate the SAXS and SANS curves for a heterogenous sample with several cyclic nanodiscs having a distinct radius. Third, some argumentation may be put forward that indicate that the heterogeneity is of dynamic nature (such as a single set of NMR peaks, the dynamics of the MD simulation).

It is correct that there may be some remaining heterogeneity even after SEC-SAXS. The study by Bibow et al. found just a single set of peaks in their NMR spectra, suggesting (i) that the two belt proteins are in an identical (average) environment, and (ii) that under those conditions the sample was relatively homogenous.

It is known that it is generally not possible to separate dispersity in size or shape from SAS data alone, as e.g. previously discussed for ellipsoidal particles (Caponetti et al., 1993). That study shows that although the scattering from ellipsoidal particles can be represented by a polydisperse distribution of spheres, the required distribution is very peculiar and unlikely.

As we have previously discussed for the slightly larger MSP1D1 discs (Skar-Gislinge et al., 2010) the nanodiscs are atypical colloid particles in the respect that their circumference is defined by the mostly fixed length of the rather unelastic and chemically well-defined MSP belts. This has the consequence that less than maximal lipid loading will be manifested in symmetry broken discs, where an elliptical shape of the lipid bilayer patch is most simple, rather than in discs of a smaller radius. It also has the consequence that a dispersity in the number of lipids embedded in the nanodiscs for a given experimental sample will, to first order, be manifested in a sample of elliptical discs of varying axis ratios. Hence, assuming circular discs of different radii is not compatible with the knowledge we have about the nanodisc system.

To investigate the consequences of a variation in the number of embedded lipids, we implemented a model for the DMPC loaded ΔH5-ΔHis discs that included a normally distributed dispersity of the number of embedded lipids (mean value of *N_lip_*=104), where the width of the Gaussian was defined by its relative standard deviation in the number of embedded lipids, *σ_lip_*, and truncated the Gaussian at +/-3 *σ_lip_*. An upper hard limit for *N_lip_* in the distribution was furthermore defined by the *N_lip_* that yielded circular and hence fully loaded discs. We chose a lower hard limit to be given by the *N_lip_* that yielded discs of axis ratios ≥ 2. We evaluated this model using the parameters optimised for the SAXS data of the ΔH5-ΔHis sample at 30°C (Table 1, fourth column) using values of *σ_lip_* of 0.01, 0.03 and 0.05 to represent increasing levels of dispersity, and used *σ_lip_* = 0.0001 to represent the monodisperse case (Figure 1—figure supplement 4). The results show that as the relative dispersity increases beyond 0.01 we obtain a less good description of our data, in particular with less pronounced oscillations around q=0.1 Å^-1^ – 0.2 Å^-1^, thus resembling the effect of an increased axis ratio of the discs.

The analysis revealed a strong dependence between *N_lip_* and the axis ratio of the lipid bilayer patch for the discs, so while the mean of the Gaussian of *N_lip_*=104 yielded an axis ratio of 1.3, then already at *N_lip_*=106, fully circular (axis ratio 1.0) and hence fully loaded discs were obtained. Conversely, an axis ratio of 2.0 was obtained already for *N_lip_*=94. As for the investigations of the axis ratio (see above), the model parameters are correlated, so that an increase of the dispersity, e.g. to σ_lip_=0.05, in part can be compensated by variation in some of the remaining model parameters. However, the re-optimized fits that could be obtained for the σ_lip_=0.05 sample were still significantly less good due to the more smeared out features than the one obtained when assuming rather monodisperse discs (σ_lip_≤0.01). This leads us to conclude that monodisperse model in terms of the number of embedded lipids does indeed provide a good description of the discs.

6) What do the authors mean with "membrane mimicking capabilities"? Just physical properties of lipids in nanodiscs as compared to liposomes?

Yes, this was poorly described and has been rephrased in the revised version. We indeed simply meant to which extent the order of the lipids resembles those in a more extended bilayer. We have updated the text to reflect this.

7) It is not clearly described how SAXS and SANS data have been scaled for data fitting. Is it just a hardware-dependent correction or does it have an underlying physical meaning? This requires a short explanation.

Briefly, our SAXS and SANS measurements are done on an absolute scale in units of cm^-1^. The data are calibrated during the experiment (SAXS data using water or BSA as secondary standards and SANS from an absolute count of the number of neutrons), so that we know the expected amount of forward scattering, i.e. *I(q=0)*, to expect for a given sample of a given concentration. We can thus use agreement between expectation and observation as a quality control measure. In contrast, un-calibrated experiments would instead treat *I(q=0)* as a free parameter. In practice, however, uncertainties in e.g. actual sample concentration means that these numbers do not match exactly, and we generally accept variations in 10%-20% for a SAXS experiment, and slightly higher numbers for a SANS experiment like that described here. We have updated the manuscript to make this clearer.

8) For comparing the NMR structure with the simulation results, have the authors considered the entire NMR structural bundle (10 structures) or just a single structure? This might have a marked impact on the chi-square value for the NMR structure.

We have performed these calculations on a single structure, because the NMR ensemble (PDB ID 2N5E) is not a “dynamic ensemble”, but instead an “uncertainty ensemble” so that the collection of 10 structures are not meant to represent the heterogeneity of the nanodics. We note, however, that all 10 structures are very similar, with typical Cα RMSDs <2Å over nearly all residues (excluding just a few poorly-defined or mobile residues at the tails).

9) Correlation of MD with amide NOES (subsection “Integrating experiments and simulations”): The authors might want to add here that the amide-amide NOEs report on the α-helical secondary structure of the individual MSPs and are not quite sensitive to slight changes in the shape of the overall MSP belt. Thus, a good correlation with these restraints is very much expected.

We agree, and mention this in the revised manuscript.

10) Subsection “Integrating experiments and simulations”, seventh paragraph: it seems like the square root(C) value for the NMR/EPR structure is missing.

This value has been added in the revised manuscript.

11) Order parameters (Figure 4B): What C-H moieties in DMPC have been used to calculate the numbers plotted in the figure? Also, it seems counter-intuitive that lipids in the rim region are more mobile than lipids in the core.

The order parameters were calculated and averaged over all C-H groups in the two alkyl chains in DMPC; this is now described more clearly in the revised manuscript.

We agree that it is not obvious that the lipids in the rim region would be more conformationally heterogeneous than those in the centre of the disc. This observation is, however, in agreement with previous simulation results (Siuda and Tieleman, 2015; Debnath and Schäfer, 2015) and we speculate that the irregular belt-protein surface makes it difficult for the lipids to be highly ordered. This is now discussed briefly in the revised manuscript.

12) It is very good that the authors make their MD data publicly available on GitHub. It would be nice if the refined ensemble (or some number of representative frames) could be made available to the scientific community also via a database, such as PDB-dev.

We have opted not to deposit the ensemble in PDB-dev, in part because the technical requirements for reformatting the data are non-trivial, and in part because it was unclear to us that PDB-dev is the right place to deposit re-weighted molecular dynamics simulations. As the reviewers note, our simulations and analyses are already available online.

13) The clustering step of the MD simulations seems quite important for the rest of the paper, and the choice of methods and parameters is not rationalized. Were other methods/parameters tested and why did the authors settle on this specific choice?

We realize that we may have not adequately described the goal and impact of clustering. In particular, the clustering is only used for visualization and to gain an overview of the ensemble, but is not part of the reweighting procedure (see e.g. the last paragraph of the subsection “Integrating experiments and simulations” in the original manuscript). Indeed, this differentiates our approach from some of the earlier work on reweighting simulations against SAXS data. Instead, in our method we reweight each frame in the original MD simulation, and then accumulate the weights over the clusters. This makes our analyses much more robust against noise that would come from clustering. For these reasons we also did not try different clustering methods. We did examine different cutoffs, and found that the chosen value, giving rise to six clusters and with the top three clusters each having >10% population, represented the data sufficiently well. We have updated the text to make it clearer that the clustering is used only for analyses and not reweighting.

14) Additionally, the reviewers had the following suggestion, but did not strictly require additional experiments to be performed: Have the authors tried to perform SANS contrast matching experiments to obtain scattering data of the protein belt and the lipid patch only, respectively? Since the NMR structure does not contain lipids, such experiments would be helpful to validate the "lipid-free" structure.

We have indeed previously attempted such experiments based on contrast variation neutron scattering, but so far without the required signal-to-noise to provide useful information. We have previously published the production and analysis of “stealth”, neutron-invisible nanodiscs for use when examining membrane proteins embedded within such discs (Maric et al., 2014). Our pilot experiments on MDP1D1 with either lipids or belt-proteins matched out, however, suggest that we need stronger neutron sources to obtain sufficient signal to provide information beyond that intrinsically available in the multi-contrast SAXS experiments.